# Bacterial Bloodstream Infections after Allogeneic Hematopoietic Stem Cell Transplantation: Etiology, Risk Factors and Outcome in a Single-Center Study

**DOI:** 10.3390/microorganisms11030742

**Published:** 2023-03-14

**Authors:** Jessica Gill, Alessandro Busca, Natascia Cinatti, Roberto Passera, Chiara Maria Dellacasa, Luisa Giaccone, Irene Dogliotti, Sara Manetta, Silvia Corcione, Francesco Giuseppe De Rosa

**Affiliations:** 1Division of Hematology, Department of Molecular Biotechnology and Health Sciences, University of Torino, A.O.U. Città della Salute e della Scienza di Torino, 10126 Turin, Italy; jessica.gill@edu.unito.it; 2Department of Oncology and Hematology, SSD Stem Cell Transplant Center, A.O.U. Città della Salute e della Scienza di Torino, 10126 Turin, Italy; 3Division of Internal Medicine, Department of Medical Sciences, University of Torino, A.O.U. Città della Salute e della Scienza di Torino, 10126 Turin, Italy; 4Department of Medical Sciences, A.O.U. Città della Salute e della Scienza di Torino, University of Torino, 10126 Turin, Italy; 5Division of Infectious Diseases, Department of Medical Sciences, A.O.U. Città della Salute e della Scienza di Torino, University of Torino, 10126 Turin, Italy

**Keywords:** bloodstream infections, allogeneic hematopoietic stem cell transplantation, multidrug-resistant bacteria, anti-bacterial prophylaxis, fluoroquinolones, antimicrobial stewardship, immunocompromised host

## Abstract

Background—Allogeneic hematopoietic stem cell transplant (allo-HSCT) recipients are subject to major risks for bacterial bloodstream infections (BSIs), including emergent multidrug-resistant (MDR) organisms, which still represent the main cause of morbidity and mortality in transplanted patients. Methods: We performed an observational, retrospective, single-center study on patients undergoing allo-HSCT between 2004 and 2020 at the Stem Cell Transplant Unit in Turin to assess the incidence, etiology, and outcomes of BSIs and to explore any risk factors for bacteriaemia. Results: We observed a total of 178 bacterial BSIs in our cohort of 563 patients, resulting in a cumulative incidence of 19.4%, 23.8%, and 28.7% at 30, 100, and 365 days, respectively. Among isolated bacteria, 50.6% were Gram positive (GPB), 41.6% were Gram negative (GNB), and 7.9% were polymicrobial infections. Moreover, BSI occurrence significantly influenced 1-year overall survival. High and very high Disease Risk Index (DRI), an haploidentical donor, and antibacterial prophylaxis were found as results as independent risk factors for bacterial BSI occurrence in multivariate analysis. Conclusions: In our experience, GNB have overwhelmed GPB, and fluoroquinolone prophylaxis has contributed to the emergence of MDR pathogens. Local resistance patterns and patients’ characteristics should therefore be considered for better management of bacteremia in patients receiving an allogeneic HSCT.

## 1. Introduction

Hematopoietic stem cell transplantation (HSCT) has been demonstrated to be a treatment strategy to cure patients with many malignant and non-malignant haematological disorders, including leukaemia, lymphomas, and aplastic anaemia [1,2].

Allogeneic HSCT (allo-HSCT) is a procedure in which recipient stem cells are first destroyed by a conditioning therapy and then restored by a donor stem cell infusion. Conditioning therapy, consisting of chemotherapy with or without total body irradiation, aims both to optimize tumor cell killing and to immunosuppress the recipient, preventing graft rejection [3].

In addition, immunosuppressive agents are essential for a long time after allo-HSCT to mitigate the Graft-versus-Host Disease (GvHD) reaction [4,5].

According to these considerations, patients undergoing HSCT are at elevated risk for severe, life-threatening infections, such as bacterial, viral, and fungal infections [6,7,8].

Indeed, the last two decades have witnessed a significant rise in bacterial bloodstream infections (BSIs) among immunocompromised patients. BSIs affect 11% to 40% of neutropenic patients [9], with an associated mortality ranging from 5% up to 60% in cases of multidrug-resistant (MDR) BSIs [10]. The epidemiology of BSI has been changing over the years. During the nineties, gram-positive bacteria (GPB) emerged as a leading cause of BSIs following the increased use of intravascular devices and the extensive use of prophylaxis with fluoroquinolones (FQ). Subsequently, this trend was followed by a gradual rise of gram-negative bacteria (GNB) and, more recently, MDR-GNB, extended-spectrum beta-lactamase (ESBL)-producing, or carbapenem-resistant (CRE) Enterobacteriaceae are becoming a critical issue in daily clinical practice [11].

Several factors might have influenced the epidemiology of BSIs. The consistent increase in grafts from alternative donors, the use of reduced-intensity preparative regimens making possible the extension of allo-HSCT to older patients and patients with relevant comorbidities once considered ineligible for the transplant, and the use of immunotherapy treatments before and after the transplant may be considered as factors potentially contributing to an increase in the risk of infections in allo-HSCT recipients [12,13,14].

Our present study aimed to estimate the incidence rates and the current clinical features of BSIs during the first year post-HSCT and to explore the impact of BSIs on HSCT outcomes. This study has been enriched with the analysis of risk factors for BSI that might potentially guide a tailored approach for allotransplanted patients.

## 2. Materials and Methods

### 2.1. Study Design and Data Collection

This observational retrospective study analyzed all consecutive episodes of BSIs occurring in adult patients who had undergone an allo-HSCT for hematologic malignancies between January 2004 and December 2020 at the Stem Cell Transplant Center, AOU Città della Salute e della Scienza of Turin.

Indications for allogeneic HSCT were hematologic malignant diseases and included: Acute Myeloid Leukemia (AML), Myelodysplastic syndromes (MDS), Acute Lymphoblastic Leukemia (ALL), Lymphomas, Chronic Myeloid Leukemia (CML), and Multiple Myeloma (MM). We excluded transplants for non-hematologic malignancies or for solid tumors. All study participants had given written informed consent for allo-HSCT and for the use of medical records for research purposes. Demographic and clinical data were retrieved from a computerized database of prospectively collected data, updated until September 2022. Patients receiving multiple allo-HSCT during the study period were censored at the time of their second or third HSCT, and data were collected independently for each single transplant.

Only BSIs occurring within the first year of transplantation were considered. If multiple bacterial infections occurred, only the first episode was examined in the analysis. Due to the retrospective observational nature of this research and according to Italian law [15], no formal approval from the local Institutional Review Board (IRB)/Independent Ethics Committee (IEC) was required.

### 2.2. Conditioning Regimens and GvHD Prophylaxis

Conditioning regimens were selected according to the patients’ characteristics. Briefly, the most frequent myeloablative conditioning (MAC) included TBI and cyclophosphamide in patients with acute lymphoblastic leukemia (ALL) and busulfan-based regimens in patients with myeloid malignancies, lymphomas, and myeloproliferative disorders. In patients over 55 years of age and/or in the presence of significant pre-transplant comorbidities, reduced-intensity (RIC)/non-myeloablative regimens were usually preferred.

GvHD prophylaxis, including cyclosporine and short-course methotrexate, was used in patients receiving grafts from matched sibling donors (MSD). In vivo T-cell depletion with anti-thymocyte globulin (ATG) was used in unrelated donor transplants (MUD) at a dose of 5–7 mg/kg over two or three days. All patients undergoing haploidentical HSCT received unmanipulated grafts and post-transplant cyclophosphamide on days +3 and +4 [16], followed by tacrolimus combined with mycophenolate mofetil since day +5.

### 2.3. Definitions

We defined neutropenia as an absolute neutrophil count (ANC) <500/mm^3^. Neutrophil engraftment was reached the first of 3 consecutive days with ANC above 500/mm^3^ [17].

BSI was defined as the isolation of a bacterial pathogen from at least 1 set of blood cultures (1 aerobic and 1 anaerobic) and then classified as GPB, GNB, or polymicrobial bacteriemia [18]. The etiological significance of the isolated pathogen was derived from clinical and microbiological assessment. Coagulase-negative Staphylococci (CNS) and other potential skin contaminants were considered significant when they were isolated in at least two blood culture sets. A polymicrobial BSI was defined by the detection of different bacterial microorganisms on the first day of a BSI episode. GNB were considered MDR if they produced extended-spectrum beta-lactamase (ESBL) or carbapenemase, and in all cases of non-response to ≥1 agent in ≥3 therapeutically relevant classes of antibiotics. Incidence data were based on the rate of BSI per the yearly number of transplants. We considered microbiologically proven bacterial growth in allo-HSCT recipients at any phase of transplant, from the start of conditioning to day +365. Based on the onset of BSIs relative to neutrophil recovery, bacterial infections were categorized as pre-engraftment (occurring within day +30) and post-engraftment (occurring between day +30 and last follow-up) BSIs. Only the first episode of bacteremia in each patient was included in the analyses.

Hematopoietic Cell Transplantation—Specific Comorbidity Index (HSCT-CI) [19] and the revised Disease Risk Index (r-DRI) [20] were used for assessing the transplant risk for each patient, respectively, for the comorbidity and the underlying disease impact.

Both acute and chronic GvHD were diagnosed based on clinical symptoms and/or biopsies according to standard criteria [21]. The severity of chronic GvHD was assessed according to National Institutes of Health criteria and graded as mild, moderate, or severe [22,23].

BSI-related mortality was defined as death within 7 days after diagnosis of BSI with the bacterial infection judged unresolved and no other ascertainable cause identified by the treating physician or from a medical record review. Non-relapse mortality (NRM) was defined as death without disease progression or relapse.

### 2.4. Supportive Care

Candidates for allograft were cared for in single rooms with positive pressure and HEPA filters. Patients were screened for microbiological contaminants with nasal, oropharyngeal, and rectal swabs and urinalysis at admission and then weekly. As all patients had an indwelling bilumen central line catheter placed at the time of admission, nursing staff guaranteed periodical dressing and site care. A central venous catheter was maintained for the first 3 months after stem cell infusion, if infection, venous thrombosis, or additional complications did not occur.

Granulocyte colony stimulating factor was not routinely administered after HSCT, except for patients grafted from haploidentical donors.

From 2004 to 2017, FQ prophylaxis with levofloxacin 500 mg orally daily was given to all patients from day +1 until neutrophil engraftment; after 2017, no universal antibacterial prophylaxis was given to patients receiving allo-HSCT. Fluconazole was used as antifungal prophylaxis from day 0 until day +75 in MRD and MUD transplants, or posaconazole if GvHD occurred, while haploidentical HSCT recipients received micafungin 50 mg/day until neutrophil engraftment, followed by fluconazole until day +75. For the first post-transplantation year, acyclovir 400 mg twice daily and cotrimoxazole 3 times weekly were provided, respectively, as anti-herpes simplex virus, anti-pneumocystis, and toxoplasma prophylaxis.

In cases of fever, defined as body temperature ≥38.3 °C once or ≥38.1 °C twice within 1 h, standard procedures were performed, including at least two sets of blood cultures (1 aerobic bottle and 1 anaerobic bottle) at two different sites and the serum β-D-Glucan antigen test. Broad-spectrum i.v. antibiotics, like piperacillin/tazobactam, were started empirically unless there was a previous infection with a resistant strain or an isolation of a pathogen sensitive to other antibiotics.

### 2.5. Statistical Analysis

Primary endpoint was the cumulative incidence (CI) for BSI (main event); competing event was the relapse/death without BSI, while alive patients were censored at the date of last contact (September 2022). The CI function was compared across groups by the Gray-test, while the competing risks regression model was used to estimate the role of risk factors on BSI occurrence by the Fine-Gray test. The following covariates were tested as BSI occurrence determinants: recipient age (>60 vs. 40–60 vs. <40 years), recipient gender (male vs. female), diagnosis (AML vs. other hematological malignancies), disease status at transplant (advanced vs. early disease), Sorror Comorbidity Index (HCT-CI, ≥3 vs. 0–2), Disease Risk Index (DRI, high/very high vs. intermediate vs. low), donor type (haploidentical vs. MUD vs. MSD), stem cell source (peripheral blood vs. bone marrow), median CD34+ doses infused/recipient weight (≥7.0 × 10^6^/kg vs. <7.0 × 10^6^/kg), median CD3+ doses infused/recipient weight (≥2.5 × 10^8^/kg vs. <2.5 × 10^8^/kg), conditioning regimen (reduced intensity vs. myeloablative), anti-thymocyte globulin (ATG) administration as GvHD prophylaxis (yes vs. no), antibiotic prophylaxis (yes vs. no), median duration of neutropenia (≥16 vs. <16 days), acute (grade II–IV vs. 0–I), and chronic GVHD (moderate and severe vs. no and mild) occurrence.

Secondary endpoint was overall survival (OS), defined as the time from transplant to death from any cause. Survival curves were estimated by the Kaplan–Meier method and compared across groups by the log-rank test. The effect on OS of the same set of covariates was analyzed by the Cox proportional hazards regression model, comparing the two arms by the Wald test, and calculating 95% confidence intervals.

Patient characteristics were tested using Fisher’s exact test for categorical variables and the Mann–Whitney and Kruskal–Wallis tests for continuous ones; continuous variables were described as median (Inter Quartile Range-IQR). All reported *p*-values were obtained by the two-sided exact method at the conventional 5% significance level. Data were analyzed as of September 2022 by R 4.2.2 (R Foundation for Statistical Computing, Vienna-A, http://www.R-project.org, accessed on 1 September 2022).

## 3. Results

### 3.1. Patients’ Characteristics

Demographic patients’ characteristics are summarized in Table 1.

During the study period, 523 patients underwent 563 allo-HSCTs and have been included in the analysis, and 36 patients had two transplants and four patients had three transplants.

Median patient age at the time of transplant was 49 years (IQR: 40–58 years), with 307 males (54.5%) and 256 females (45.5%).

Acute leukemia was the most common indication for allo-HSCT (*n* = 403, 71.6%). Overall, 67.5% of patients were in complete remission (CR) at transplant (282/560, 50.4% in first CR; 96/560, 17.1% in second CR), and 113 patients (20.9%) had a high (92/541) or very high (21/541) r-DRI.

In all, 191 patients (34%) received HLA-identical sibling transplants, and 371 (66%) received grafts from alternative donors (*n* = 284 MUD, *n* = 87 haploidentical donors). The graft source was peripheral blood in 81.5% of the patients. A myeloablative conditioning (MAC) regimen was used in 69.4% of the transplants. The median time to neutrophil engraftment was 16 days (IQR: 14–18 days). During the aplastic phase, most patients (381/534, or 71.3%), received antibacterial prophylaxis with levofloxacin.

### 3.2. Incidence and Clinical Characteristics of BSIs

Overall, 178 (31.6%) patients were diagnosed with bacterial BSIs during the post-HSCT follow-up, with a cumulative incidence (CI) of 19.4%, 23.8%, and 28.7% at days +30, +100, and +365, respectively (Figure 1).

Most bacteriaemia occurred in the pre-engraftment phase (109/178—61.2%), with 58.4% (104/178) of patients neutropenic at the time of infection; the remaining 38.8% developed in the post-engraftment period (69/178). Among pre-engraftment BSIs, 23 infections presented during conditioning chemotherapy and two at day 0.

The median time of onset of BSI was 9 days after the transplant (IQR: 6–81 days).

According to different donor type, the 1-year CI of BSI was 17.3% for MSD, 31.4% for MUD, and 45% for haploidentical donor transplants (Figure 2A).

Among patients with different DRIs, the 1-year CI was 9.8% for low DRI, 27.9% for intermediate DRI, and 40.9% for high and very high DRI transplant recipients (Figure 2B).

The 1-year CI of BSI was 22.6% and 40.1%, respectively, in patients who received or did not receive antibacterial prophylaxis.

Patients with acute leukemia had the highest rates of GPB, GNB, and polymicrobial infections (82%, 84%, and 79%, respectively) and MDR-GNB (95%). Similarly, patients receiving grafts from alternative donors had the highest rate of BSI (GPB 79%, GNB 77%, polymicrobial 71%, MDR-GNB 85%), as did patients receiving PBSC as a stem cell source (GPB 67%, GNB 86%, polymicrobial 71%, MDR-GNB 90%) and patients on myeloablative regimens (GPB 74%, GNB 76%, polymicrobial 71%, MDR-GNB 65%).

### 3.3. Etiology of BSIs

Table 2 details the etiology of BSIs according to the different time onsets from HSCT. Gram-negative bacteria (GNB) were isolated in 74/178 patients (41.6%) with BSI, Gram-positive bacteria (GPB) in 90/178 patients (50.6%) and polymicrobial in 14/178 patients (7.9%).

Among gram-positive BSI, the most frequent isolates of pathogens were coagulase-negative *Staphylococci* (*n* = 66; 73%), followed by *Streptococcus mitis* (*n* = 8; 9%), *Enterococcus* spp. (*n* = 8; 9%), and *Staphylococcus aureus* (*n* = 3; 3%). Among gram-negative BSI, *Enterobacteriaceae* were the most frequent isolates (61/74, 82%), especially *Escherichia coli* (*n* = 39; 53%), followed by *Klebsiella pneumoniae* (*n* = 12; 16%), and *Pseudomonas aeruginosa* (*n* = 10; 13%). Polymicrobial BSIs were mainly due to the association of GNB (*Escherichia coli* or *Klebsiella pneumoniae*) and other bacteria.

As shown in Figure 3, a sharp decrease in the rates of GPB bacteremia was observed during the study period, while the rates of BSI caused by GNB increased significantly in recent years.

Overall, 20 MDR-GNBs have been detected (20/74, 27%), occurring mainly during the pre-engraftment phase (13/20, 65%).

### 3.4. Drug Resistance

Antibiograms were performed in all patients with isolates from blood cultures. Figure 4 summarizes the evolution of susceptibility patterns of main GNB isolates over the study timeframe. The main discriminating factor was the withdrawal of FQ prophylaxis in 2017. Since then, we observed a decrease in ESBL isolates; instead, the rates of carbapenem-resistant Enterobacteriaceae (CRE) and MDR-Pseudomonas aeruginosa remained roughly stable (4/29, 14%, and 2/5, 40%, respectively, in the last time period), although the number of isolates was consistently low.

### 3.5. Risk Factors for Bacterial BSI

Table 3 summarizes univariate and multivariate competing risk regression models for BSI occurrence. AML diagnosis, high and very high DRI before HSCT, receipt of grafts from haploidentical donors, the number of CD3+ cells infused, and the absence of antibacterial prophylaxis were associated with a higher risk for bacterial BSI in univariate analyses (SDHR 1.69, *p* = 0.001; SDHR 1.91, *p* < 0.001; SDHR 1.76, *p* < 0.001; SDHR 0.68, *p* = 0.020; SDHR 0.51, *p* < 0.001, respectively). High and very high DRI (SDHR 1.70, IC 1.26–2.31; *p* < 0.001), an haploidentical donor (SDHR 1.51, IC 1.17–1.93; *p* = 0.001) and the antibacterial prophylaxis (SDHR 0.58, IC 0.40–0.84; *p* = 0.004) resulted to be associated with a higher risk of BSI in multivariate analysis.

### 3.6. Outcome and Mortality

In our cohort, 295 (52.4%) patients were alive after allo-HSCT, with a median follow-up of 6.70 years (range 50 days–17.3 years). Median OS was 8.3 years. Two hundred and sixty-eight patients (47.6%) died, in particular 44.7% of patients without BSI (172/385), and 53.9% of patients with BSI (96/178).

The major cause of death was relapse (168/268—62.7%), while one third of the patients died of transplant-related complications (100/268—37.3%): infectious complications accounted for 52% of deaths, followed by GvHD (26%), a second tumor (16%), and other causes (6%).

BSI-related mortality rate was 10.3%, with a median follow-up of 109 days. BSI-related mortality was not significantly different between patients who did or did not receive FQ prophylaxis (7.7% vs. 4.6%).

As expected, the occurrence of BSI had a significant impact on overall survival at 1 year (BSI 55.5% vs. non-BSI 77.4%) (Figure 5) and median OS (BSI 1.3 years vs. non-BSI 10.3 years) (*p* < 0.001).

## 4. Discussion

BSIs continue to occur at an elevated rate among HSCT recipients, leading to increased length of hospitalization, intensive care admission, and mortality. The value of the present study is a large number of adult HSCT recipients captured between 2004 and 2020, providing essential contemporary data to inform the management of infectious complications in this cohort of hematologic patients. Subjects included in this study were at high risk of BSIs: over two-thirds of the patients were affected by acute leukemia, received myeloablative regimens, and were grafted from alternative donors, including unrelated and haploidentical donors. It should be emphasized that the most recent studies did not show a significant difference in the rate of BSI between haploidentical grafts receiving post-transplant cyclophosphamide as GVHD prophylaxis and patients undergoing HSCT from a matched sibling or unrelated donors, with a reported incidence ranging between 30 and 40% [24].

Our study documented a cumulative incidence of BSI of 19% and 28%, respectively, at 30 days and 1-year post-HSCT. Our results compare favorably with those reported by large studies, which range from 21% up to 55% [25,26,27,28]. According to the most recent epidemiological data, we observed a progressive increase in GNB over GPB, with a preponderance of *Enterobacteriaceae* and non-fermenter bacteria and a peak of MDR pathogens in the 2013–2016 period, as reported also in other Italian centers [29,30]. These findings emphasize the importance to corroborate the awareness of local epidemiology.

In multivariate analysis, three factors emerged as significantly associated with the occurrence of BSIs. Patients receiving grafts from a matched sibling donor had a 17% incidence of BSIs, significantly lower as compared to MUD (31%), and haploidentical (45%). Reasonably, the gap might result from the different immunosuppressive treatments adopted for GVHD prophylaxis in the three groups of HSCT: calcineurin inhibitor and methotrexate for matched sibling donor HSCT, ATG for MUD, and post-transplant cyclophosphamide for haploidentical grafts. In addition, the high rate of GVHD reported in patients receiving grafts from alternative donors might be considered a factor potentially contributing to the risk of BSI.

The DRI incorporates disease, disease status at the time of HSCT, and cytogenetic markers in AML and MDS, all of which are critical determinants of transplantation outcome. This index applies regardless of age, preparative regimen, donor type, and graft source. In our study, high/very high DRI is associated with the risk of BSI. This finding should not be considered unexpected since the DRI score mirrors the intensity of the chemotherapy treatments administered before the transplant, which in turn may contribute to a decline in the patients’ fitness and treatment tolerance.

The benefit of FQ prophylaxis in patients receiving HSCT is not standardized or consistent [31,32,33]. The results of a survey of Blood and Marrow Transplant Clinical Trials Network centers evaluating antibacterial antibiotic practices among 41 HSCT centers showed that FQ prophylaxis was used in 75% of the centers [34]. Scattered studies have shown that antibacterial prophylaxis with FQ is associated with a reduced rate of BSI caused by GNB; however, the drawback of resistance has always been a concern [35]. In addition, antibacterial prophylaxis may contribute to gut microbiota perturbation with a decrease in microbiome diversity, which is associated with the risk of GVHD and NRM [36]. Based on these considerations, we decided to omit FQ prophylaxis since 2017: the incidence of BSI was 60% among patients who received prophylaxis and 40% in those who did not. On the other hand, this strategy resulted in a reduction in BSI caused by ESBL *Enterobacteriaceae* and progressive recovery of antibiotic sensibility to FQ. Of note, the mortality correlated to BSI was roughly low (10%), consistent with the data reported in the literature [28,37,38], even considering that 26% of GNB BSI were caused by MDR bacteria that are characterized by high mortality [39,40,41]. More importantly, the crude infection-related mortality was compared between patients who received and did not receive FQ prophylaxis (7.7% vs. 4.6%). Concern for an increased risk of *Clostridium difficile* infections is another argument for avoiding the use of FQ; however, this infectious complication was not captured by our database.

There are a few limitations to this study. This was a retrospective and single-center study with a geographically defined cohort, and therefore our findings may not be generalizable to other areas.

Nevertheless, our results unveil factors potentially affecting the risk of BSIs may be relevant to determining successful interventions aiming to guide infection control and antimicrobial stewardship and eventually improving the outcome of patients within vulnerable populations, such as those undergoing allo-HSCT. For this reason, we routinely perform infection surveillance with rectal swabs on patients admitted for allo-HSCT; there is an ongoing analysis evaluating the potential role of the rectal swab in predicting the subsequent development of MDR BSI. Moreover, there are two important therapeutic approaches that we have adopted: de-escalation therapy in patients with non-MDR BSIs and carbapenem-sparing therapy.

## 5. Conclusions

In conclusion, our study confirmed a notorious change in the epidemiology of bacterial infections in allo-HSCT recipients, with a progressive increase in GNB over GPB. Recently, an emerging role for MDR-BSI has been revealed, probably favored by decades of FQ prophylaxis in various HSCT settings, including ours. A high and very high DRI, an haploidentical donor, and antibacterial prophylaxis with FQ have been identified as independent risk factors of bacterial BSIs. These results aim to contribute to better BSI management with a major effort toward antimicrobial stewardship.

## Figures and Tables

**Figure 1 microorganisms-11-00742-f001:**
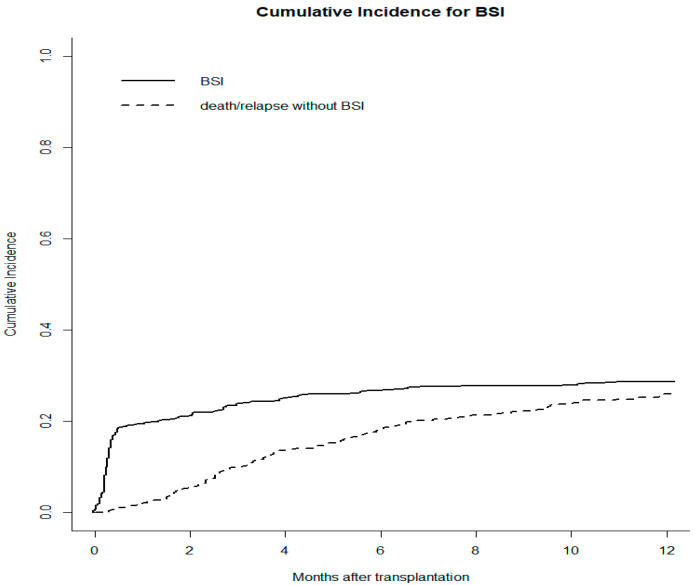
The cumulative incidence rate of bacterial bloodstream infections (BSIs) is 19.4%, 23.8%, and 28.7% at 1, 3 and, 12 months after transplant for the whole cohort. Death or relapse without BSIs is considered the competing event.

**Figure 2 microorganisms-11-00742-f002:**
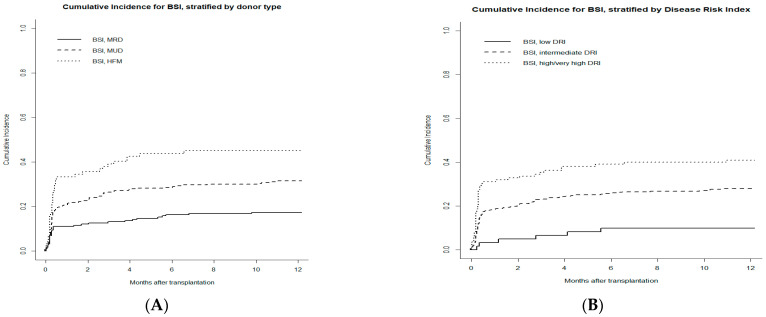
Cumulative incidence rate of bacterial bloodstream infections (BSIs) at 1 year stratified for: (**A**) donor type and (**B**) DRI. Death or relapse without BSIs is considered the competing event.

**Figure 3 microorganisms-11-00742-f003:**
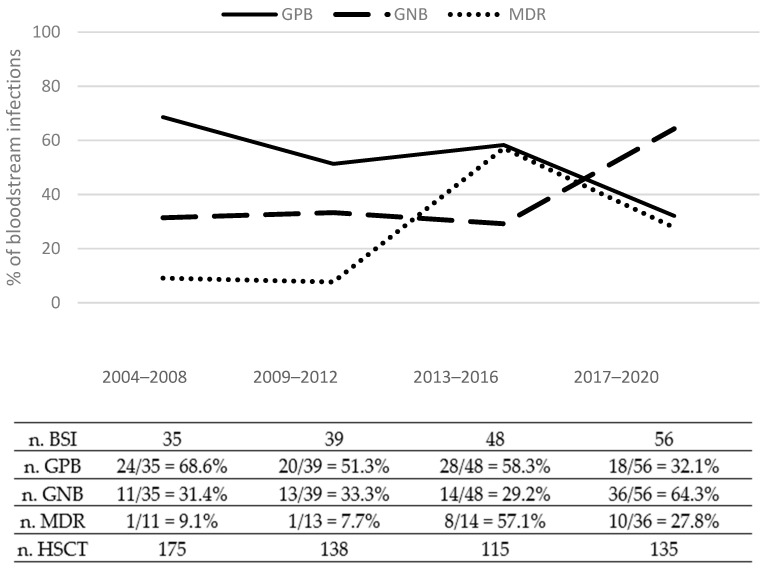
Trend of GPB, GNB, and MDR bacteria from 2004 to 2020 in our cohort. Vertical axis represents the percentage of type of infection on total BSI; horizontal axis indicates the four time periods analyzed. The table below shows absolute numbers. According to Fisher’s exact test, GPB over GNB ratio significantly changed over time (*p* < 0.001).

**Figure 4 microorganisms-11-00742-f004:**
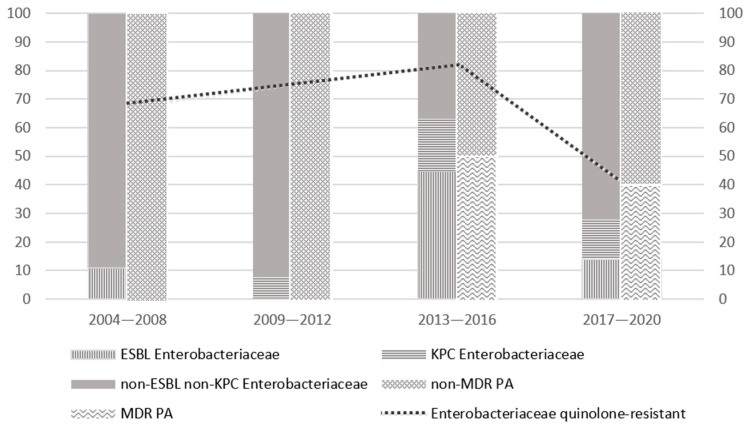
Susceptibility patterns of main GNB isolates on blood cultures and their evolution over time, in particular for *Enterobacteriaceae* (ESBL, KPC, non-ESBL non-KPC) and *Pseudomonas Aeruginosa* (MDR and non-MDR), as explicated by the legend below.

**Figure 5 microorganisms-11-00742-f005:**
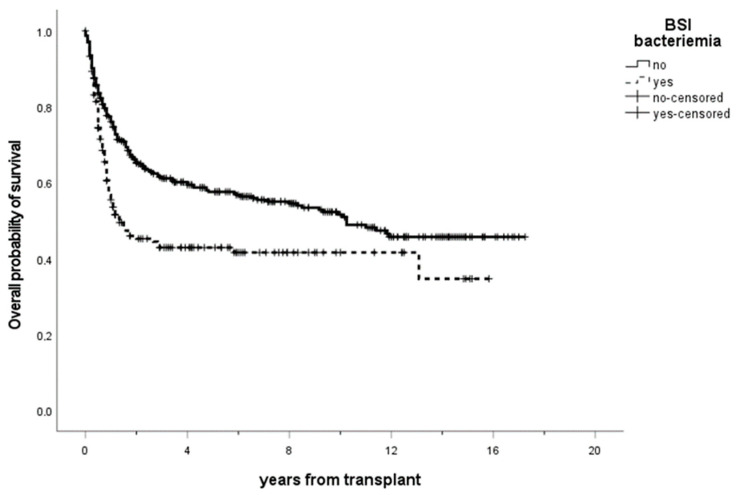
Overall survival of patients receiving allogeneic transplants with a diagnosis of BSI (dashed line) or not (solid line). The 1-year overall survival was 46.1% in BSI positive patients vs. 55.3% in BSI negative patients, with a statistically significant difference (*p* < 0.001). Ticks on probability lines indicate dates of censoring at the last follow-up.

**Table 1 microorganisms-11-00742-t001:** Main patients’ and transplants’ characteristics of the whole cohort and stratified by BSI.

Characteristics	All Patients	BSI Neg	BSI Pos
Number of patients	563	385	178
Age at transplant, median (IQR), years	49 (40–58)	49 (39–58)	49 (40–58)
Gender			
Male	307 (54.5%)	202 (52.5%)	105 (59%)
Female	256 (45.5%)	183 (47.5%)	73 (41.0%)
Underlying disease			
AML/MDS	319 (56.7%)	199 (51.7%)	120 (67.4%)
ALL	84 (14.9%)	57 (14.8%)	27 (15.2%)
HL/NHL	101 (17.9%)	80 (20.8%)	21 (11.8%)
MPN/LMMC	49 (8.7%)	39 (10.1%)	10 (5.6%)
MM	10 (1.8%)	10 (2.6%)	0 (0%)
Disease status at transplant			
CR	282 (50.4%)	194 (50.7%)	88 (49.7%)
CR2	96 (17.1%)	73 (19.1%)	23 (13%)
PIF/relapse	182 (32.5%)	116 (30.3%)	66 (37.3%)
HCT-CI			
Low/intermediate (0–2)	236 (72.6%)	151 (73.7%)	85 (70.8%)
High (>3)	89 (27.4%)	54 (26.3%)	35 (29.2%)
DRI			
Low	61 (11.3%)	54 (14.6%)	7 (4.1%)
Intermediate	367 (67.8%)	255 (68.7%)	112 (65.9%)
High	92 (17.0%)	52 (14.0%)	40 (23.5%)
Very high	21 (3.9%)	10 (2.7%)	11 (6.5%)
Donor type			
MRD	191 (34%)	151 (39.3%)	40 (22.5%)
MUD	284 (50.5%)	188 (49%)	96 (53.9%)
Haploidentical ^a^	87 (15.5%)	45 (11.7%)	42 (24%)
Stem cell source			
PBSC	458 (81.5%)	323 (84.1%)	135 (75.8%)
BM	96 (17.1%)	56 (14.6%)	40 (22.5%)
CB	8 (1.4%)	5 (1.3%)	3 (1.7%)
Number of CD34+ cells infused, median (IQR), ×10^6^/kg	7.0 (5.4–9.1)	7.2 (5.7–9.1)	6.6 (5.2–8.6)
Number of CD3+ cells infused, median (IQR), ×10^8^/kg	2.5 (1.5–3.3)	2.6 (1.7–3.6)	2.2 (1.2–3.1)
Conditioning regimen			
MAC	390 (69.4%)	258 (67.2%)	132 (74.2%)
RIC	172 (30.6%)	126 (32.8%)	46 (25.8%)
GvHD prophylaxis			
ATG	280 (50.4%)	186 (48.9%)	94 (53.4%)
other	276 (49.6%)	194 (51.1%)	82 (46.6%)
Time to engraftment ^b^, median (IQR), days	16 (14–18)	16 (14–18)	16 (14–18)
FQ prophylaxis			
No	153 (28.7%)	88 (23.7%)	65 (40.1%)
Yes	381 (71.3%)	284 (76.3%)	97 (59.9%)
Acute GvHD			
0–I	175 (65.8%)	108 (65.9%)	67 (65.7%)
II–IV	91 (34.2%)	56 (34.1%)	35 (34.3%)
Chronic GvHD			
absent/mild	166 (65.6%)	102 (66.7%)	64 (64%)
moderate/severe	87 (34.4%)	51 (33.3%)	36 (36%)
Overall Survival			
Alive	295 (52.4%)	213 (55.3%)	82 (46.1%)
Dead	268 (47.6%)	172 (44.7%)	96 (53.9%)

^a^ Including five patients who received HSCT from one antigen-mismatched related donor. ^b^ Engraftment is defined as the first of three days with neutrophils >0.5 × 10^9^/L after stem cell reinfusion. Abbreviations: BSI bloodstream infection, AML acute myeloid leukemia, MDS myelodysplastic syndromes, ALL acute lymphoblastic leukemia, HL Hodgkin lymphoma, NHL non-Hodgkin lymphoma, MPN myeloproliferative neoplasms, LMMC chronic myelomonocytic leukemia, HCT-CI hematopoietic cell transplantation-comorbidity index, DRI disease risk index, CR first complete remission, CR2 second complete remission, PIF primary induction failure, MRD matched related donor, MUD matched unrelated donor, PBSC peripheral blood stem cell, BM bone marrow, CB cord blood, IQR inter quartile range, MAC myeloablative conditioning, RIC reduced intensity conditioning, ATG anti-thymocyte globulin, GvHD graft versus host disease, and FQ fluoroquinolone.

**Table 2 microorganisms-11-00742-t002:** Microbiological etiology of BSIs diagnosed.

	Whole Cohort (*n* = 178)	Pre-Engraftment (day 0–30) (*n* = 109)	Post-Engraftment (day ≥ 31) (*n* = 69)
**Gram-positive Bacteria**	**90**	**52**	**38**
Coagulase-negative *staphylococci* ^a^	66	34	32
*Staphylococcus aureus* (n. MRSA)	3 (2 MRSA)	1 (1 MRSA)	2 (1 MRSA)
*Streptococci mitis*	8	7	1
Other streptococci	4	2	2
*Enterococcus* (n. VRE)	8 (1 VRE)	7 (1 VRE)	1 0
*Corynebacterium* spp.	1	1	-
**Gram-negative Bacteria**	**74**	**47**	**27**
*Escherichia coli* (n. ESBL)	39 (7 ESBL)	30 (6 ESBL)	9 (1 ESBL)
*Klebsiella* (n. ESBL) (n. KPC)	12 (2 ESBL) (7 KPC)	7 (0 ESBL) (5 KPC)	5 (2 ESBL) (2 KPC)
*Enterobacter* (n. ESBL)	4 (1 ESBL)	3 (1 ESBL)	1
*Stenotrophomonas maltophilia*	3	-	3
*Citrobacter*	1	1	-
*Proteus mirabilis*	2	1	1
*Pseudomonas aeruginosa* (n. MDR)	10 (3 MDR)	3 (1 MDR)	7 (2 MDR)
*Fusobacterium*	2	2	-
*Bacteroides*	1	-	1
**Polymicrobial BSI**	**14**	**10**	**4**
**MDR-gram-negative bacteria**	**20**	**13**	**7**

Abbreviations: MRSA Methicillin-resistant *Staphylococcus aureus*, spp. species, VRE Vancomycin-resistant enterococci, ESBL Extended Spectrum Beta-Lactamase, KPC Carbapenemase-producing *Klebsiella* pneumoniae, MDR Multidrug-resistant, BSI bloodstream infections. ^a^ Including: 43 Methicillin-resistant *Staphylococcus epidermidis*, 8 Methicillin-sensitive *Staphylococcus epidermidis*, 8 *Staphylococcus haemolyticus*, 6 *Staphylococcus hominis*, 1 *Staphylococcus capitis*.

**Table 3 microorganisms-11-00742-t003:** Univariate and multivariate competing risk regression for bacterial BSI after allo-HSCT.

	**Univariate Analyses**		**Multivariate Analysis**	
	**SDHR (95% CI)**	** *p* **	**SDHR (95% CI)**	** *p* **
Age (>60 vs. 40–60 vs. <40 years)	1.02 (0.78–1.22)	0.850	-	-
Gender (male vs. female)	1.27 (0.93–1.74)	0.130	-	-
Underlying disease (AML vs. other)	1.69 (1.23–2.34)	0.001	1.31 (0.91–1.90)	0.150
Disease status at transplant (advanced vs. early disease)	1.08 (0.79–1.47)	0.620	-	-
HCT-CI (≥3 vs. 0–2)	1.11 (0.74–1.68)	0.610	-	-
DRI (very high + high vs. intermediate vs. low)	1.91 (1.46–2.49)	<0.001	1.70 (1.26–2.31)	<0.001
Donor type (Haploidentical vs. MUD vs. MRD)	1.76 (1.41–2.20)	<0.001	1.51 (1.17–1.93)	0.001
Stem cell source (PBSC vs. BM)	0.70 (0.48–1.01)	0.055	0.80 (0.49–1.29)	0.360
Number of CD34+ cells infused (over vs. under median), 10^6^/kg	0.77 (0.57–1.06)	0.110	-	-
Number of CD3+ cells infused (over vs. under median), 10^8^/kg	0.68 (0.49–0.94)	0.020	0.15 (0.55–1.21)	0.300
Conditioning regimen (RIC vs. MAC)	0.71 (0.50–1.02)	0.061	0.87 (0.57–1.32)	0.640
GvHD prophylaxis (ATG vs. other)	1.24 (0.91–1.69)	0.170	-	-
Antibacterial prophylaxis (yes vs. no)	0.51 (0.37–0.70)	<0.001	0.58 (0.40–0.84)	0.004
Time to engraftment (over vs. under median), days	0.97 (0.70–1.33)	0.840	-	-
Acute GvHD (II–IV vs. 0–I)	1.01 (0.67–1.53)	0.950	-	-
Chronic GvHD (moderate-severe vs. no and mild)	1.13 (0.75–1.70)	0.550	-	-

Abbreviations: BSI bloodstream infection, HSCT hematopoietic stem cell transplant, SDHR sub-distribution hazard ratio, CI confidence interval, AML acute myeloid leukemia, HCT-CI hematopoietic cell transplantation comorbidity index, DRI disease risk index, MUD matched unrelated donor, MRD matched related donor, PBSC peripheral blood stem cell, BM bone marrow, RIC reduced intensity conditioning, MAC myeloablative conditioning, GvHD graft-versus-host disease, ATG anti-thymocyte globulin.

## Data Availability

The data presented in this study are available on request from the corresponding author. The data are not publicly available due to privacy restrictions.

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
