# Peer review of "Bacterial Bloodstream Infections after Allogeneic Hematopoietic Stem Cell Transplantation: Etiology, Risk Factors and Outcome in a Single-Center Study"

_microorganisms, 2023, doi:10.3390/microorganisms11030742_

Round 1
Reviewer 1 Report
The article “Bacterial bloodstream infections after allogeneic hematopoietic stem cell transplantation: etiology, risk factors and outcome in a single-center study” by Jessica Gill and Alessandro Busca et al. provides epidemiological data on bloodstream infections (BSI) among allogeneic hematopoietic stem cell transplant (allo-HSCT) recipients in single stem cell transplant center in Turin, Italy.
General remark:
The study covers a relevant topic and the English language and data presentation is clear and comprehensible in most parts. I have some comments that may help to improve the manuscript.
Specific comments:
- Throughout the manuscript: Names of bacteria (genus and species) should be written in italics.
- Throughout the manuscript: I suggest an in-depth proof reading of the manuscript. There are many minor grammatical and spelling errors.
- Results: “During the aplastic phase, most patients (381/534, 71.3%) received antibacterial prophylaxis with levofloxacin.” Over the 16-year study period, were there any changes in antibacterial prophylaxis regimes at this hospital (length, dose, etc.)?
- Results: “The characteristics of patients with and without BSIs are reported in Table 1.” (line 244). This sentence is not necessary and a repetition of line 193.
- Results: “As shown in Figure 3, a sharp decrease in the rates of GPB bacteremia was observed during the study period, while the rates of BSI caused by GNB increased significantly in recent years.” (lines 262-264). Given the relatively low number of samples per group and time period, the authors should consider performing statistical time trend analyses to confirm whether any visible increase or decrease is actually statistically significant.
- Figure 3: The authors should decide whether they display the absolute numbers or the percentage. Right now, it is a mix of absolute numbers displayed (all-HSCT group) and percentages (e.g. MDR group).
- Results line 280: “isolated” probably should be “isolated pathogen”?
- Results line 282: Please use “the” before “main discriminating factor”.
- Figure 4: There is no Figure 4 in the manuscript…
- Results “Figure 4. Susceptibility patterns of main GNB isolates on blood cultures and their evolution over time, in particular for Enterobacteriaceae (ESBL, KPC, non-ESBL non-KPC) and Pseudomonas Aeruginosa (MDR and non-MDR), as explicated by the legend below. ESBL extended spectrum beta-lactamase, KPC Klebsiella pneumoniae carbapenemase-producing bacteria, MDR multi-drug resistance.” (lines 287 – 291). Please revise this sentence. I do not understand what it actually means.
- Results: The “univariate and multivariate competing risk regression models” (Table 3) are not explained in the Method section. The authors may consider explaining the meaning of “sub-distribution hazard ratio” (SDHR) briefly in the legend of Table 3. For example, what does a SDHR of 1.11 (HCT-CI (≥3 vs 0-2)) actually mean? The SDHR is not too common in the field of infectious disease epidemiology. Moreover, there is a relatively large number of covariables used in the models. How were they chosen? Were there any a priori hypotheses? Moreover, given the relatively small sample number, did the authors check for the statistical power of their analyses?
- Discussion: The authors should compare their results (e.g. the etiology of BSI) with other studies on BSI among (allo-HSCT) recipients. For example, is the bacterial spectrum and AMR profile different compared to other stem cell transplant centers in other regions? If yes, why?
- Discussion: “Nevertheless, our results unveiling factors potentially affecting the risk of BSI may be relevant to determine successful interventions aiming to guide infection control and antimicrobial stewardship and eventually to improve the outcome of a vulnerable population of patients such as those undergoing allo-HSCT.” (lines 385-387). What interventions are thinkable? Which results exactly can lead to which IPC and/or ABS interventions?
Author Response
Thank you for the comments and the suggestions. We hope our revised article could response to your requests.
Reviewer 2 Report
The study by Gill et al., is a retrospective study analysing blood stream infections within a transplant setting at a single centre in Italy. The overall conclusions include 1) gram negative bacteria infections are more than that of gram negative, 2) blood stream infections had a significant impact on over all survival at 1 year post transplants, 3) haplo-identical donors and prophylaxis are main risk factors for bacterial septicemia (in multivariate analysis), 4) prophylaxis resulted in the emergence of multi drug resistance pathogens.
Major comments:
I would recommend the authors to tone down their conclusion on the emergence of MDR pathogens from the use of FQ. There isn't clear evidence to make these claims. FQ was used between 2007-13 and the peak of MDR occured in 2013-16 period. Can the authors please explain how they make this claim
The researchers should explain why they see a peak of MDR infections between 2013-2016 period and whether this is nosocomial in nature. Also does this correlate with infections within other surgical wards.
I would also significantly reduce the number of abbreviations used within the manuscript to make this topic more easy to read and comprehend. The reviewer finds that the readers often have to go back and look for abbreviations while reading the manuscript
In table 1, the researchers should provide the rage of the age of the patients as supposed to inter-quartile range.
The researchers should provide their findings in relation to similar studies done around the world.
Author Response

(The authors gave the same response as above.)
